# Optimization of ultrasonic-assisted extraction of polysaccharides and triterpenoids from the medicinal mushroom *Ganoderma lucidum* and evaluation of their in vitro antioxidant capacities

**Shizhong Zheng[1], Weirui Zhang[1,2], Shengrong Liu**📐[1] *

**1** College of Life Science, Ningde Normal University, Ningde City, Fujian, China, **2** Fujian Higher Education Research Center for Local Biological Resources in Ningde, Ningde City, Fujian, China

* fjhost@163.com

## Abstract

*Ganoderma lucidum* (Fr.) Krast, commonly known as "Lingzhi" in Chinese, is a medicinal mushroom that is rich in biologically active substances. Polysaccharides and triterpenoids are the two major components responsible for the bioactivity of this fungus. In the present study, the ultrasonic-assisted co-extraction (UACE) of polysaccharides and triterpenoids from *G. lucidum* was optimized using response surface methodology with a desirability function, with the equal importance for the two components. Following single factor experiments, the optimal conditions were determine as ultrasonic power of 210 W, extraction temperature of 80C, ratio of liquid to solid of 50 mL/g, and 100 min extraction time, using aqueous ethanol (50%, v/v) as the extracting solvent. Under the optimal conditions, the extraction yields of polysaccharides and triterpenoids reached 0.63% and 0.38%, respectively. On the basis of the scavenging capacity of 2,2-diphenyl-1-picrylhydrazyl and evaluation of reducing power, the antioxidant capacities of the polysaccharides obtained by optimal UACE process were higher than those of polysaccharides extracted using traditional hot water extraction, whereas the triterpenoid-rich extracts showed antioxidant activities similar to those obtained using the ethanol maceration method. The present study is the first report on the simultaneous extraction of polysaccharides and triterpenoids from *G. lucidum*. The developed UACE process could be useful in preparation of a polysaccharide- and triterpenoid-rich ingredient that holds great promise for application in the *Ganoderma* industry.

## Introduction

*Ganoderma lucidum* (Fr.) Krast belonging to the family of Polyporaceae, commonly known as "Lingzhi" in Chinese and "Reishi" in Japanese, is one of the most highly prized medicinal mushrooms. Its fruiting bodies have been used as a folklore remedy to treat various illnesses and promote longevity for thousands of years in many oriental countries like China and Korea

**Data Availability Statement:** All relevant data are within the paper and its Supporting Information files.

**Funding:** The research was supported by the Key Cultivating Program (No. 2018ZDK01) of Ningde Normal University and Ningde Science and Technology Plan Project (20190013).

**Competing interests:** The authors have declared that no competing interests exist.

[1]. In recent years, there has been a growing worldwide interest in utilizing *G. lucidum* in the biomedical industry due to its pharmacological properties. Currently, a variety of *G. lucidum* products such as slices, tea, capsules, tablets, and drinks are available on the market, and commonly consumed in many parts of the world [2].

*G. lucidum* in the form of fruiting bodies, mycelium, and spores contains a broad range of bioactive compounds, including polysaccharides, triterpenoids, phenols, steroids, lectin, amino acids, nucleosides, and nucleotides [3]. Among these, the polysaccharides and triterpenoids are receiving special attention from researchers in the field of biomedicine. Several polysaccharides such as β-D-glucans and glycoprotein are generally recognized as having the most noticeable biological activity such as antitumor, antioxidant and immunomodulatory effects among various *Ganoderma* polysaccharides [4, 5]. The triterpenoids from this fungus exhibit structural diversity, and many of them such as ganoderic acids Me and T (a kind of highly oxygenated lanostane-type triterpenoids) have been proven to have potent antitumor activity [6]. In this context, much effort has been paid to the polysaccharides and triterpenoids of *G. lucidum*, including extraction techniques [7, 8], their pharmacological activities and mechanisms of action [9, 10], and fermentative production [11, 12].

Polysaccharides are generally prepared by traditional hot water extraction due to its easy operation and low instrument input [13]. However, it also has many shortcomings such as low extraction yield and long extraction duration, as well as requirement of high temperature [14]. To overcome these problems, a variety of novel techniques, including pressurized liquid extraction [15], ultrasound-assisted extraction (UAE) [16], and hydrothermal extraction [17], have been developed in extracting *G. lucidum* polysaccharides. For the extraction of triterpenoids, maceration as a conventional method is adopted widely using organic solvents as an extraction solvent [18]. A novel technique of microwave irradiation was shown to be effective for the extraction of triterpenoids from this fungus [7].

In recent years, the simultaneous extraction of multiple biologically active components from plants has become increasingly popular in order to improve extraction efficacy and reduce the number of extraction steps [19, 20]. However, there have been no available reports on the simultaneous extraction of multiple objective constituents from mushrooms to date. Most of previous studies pertaining to the extraction of polysaccharides and triterpenoids from *G. lucidum* have mainly focused on the extraction yield of one or the other [7, 8]. Hence, the significant loss of either polysaccharides or triterpenoids occurred during the extraction, and this would lead to a waste of resources. To our knowledge, the simultaneous extraction of polysaccharides and triterpenoids in *G. lucidum* has not yet been reported to date. In the present study, to better utilize *G. lucidum*, ultrasound technology was examined for the co-extraction of polysaccharides and triterpenoids, and an optimization study is essential.

Response surface methodology (RSM), a collection of mathematical and statistical techniques, is widely used for optimizing complex processes [21, 22]. The major advantage of RSM is the reduced number of experimental trials needed to evaluate multiple parameters and their interactions. RSM has been employed for optimizing the extraction of *G. lucidum* polysaccharides and ganoderic acids [8, 23] and triterpenoids from the medicinal fungus *Sanghuangporus sanghuang* [24]. In some optimization studies, multiple desirable response variables are commonly required. As desirability functions can transform two or more responses with different relative importances into a single objective function; thus, it is a desirable approach for the optimization of a multiple-response process [25]. It was used in tandem with statistical experimental designs for optimization of fermentation conditions [26], and extraction processes [27].

In the present study, the ultrasonic-assisted co-extraction (UACE) of polysaccharides and triterpenoids was evaluated using ultrasound technology as an attempt to better utilize *G.*

*lucidum*. The extraction parameters were optimized using RSM with a desirability function by setting the equal importance for the two components. Furthermore, to provide a scientific basis for better utilization of polysaccharides and triterpenoids from the optimal UACE, their in vitro antioxidant activities were investigated and compared with those of polysaccharides and triterpenoids obtained using conventional extraction approaches. The UACE process developed here could be economically useful and convenient in the preparation of a polysaccharide- and triterpenoid-rich ingredient that holds great promise for the *Ganoderma* industry.

## Materials and methods

### Materials and reagents

The dried *G. lucidum* was obtained from a local pharmacy (Kang-Bai-Jia Medical Chain Store, Fujian, China). 2,2-diphenyl-1-picrylhydrazyl (DPPH), ursolic acid, ferric chloride, and trichloroacetic acid were products of Sigma-Aldrich Co. (St. Louis, MO, USA). All other chemical reagents used were of analytical grade.

### Hot water extraction of polysaccharides

The dried *G. lucidum* was grounded in a grinder (DFY-500, Zhejiang Linda Co. Ltd., China) to a fine powder (40 mesh). The fine powder sample was pretreated with 80% ethanol to remove some colored materials, oligosaccharides, and some small molecule materials [28], and then dried at 40C in an oven. The treated powder (1 g) was mixed with 40 mL of deionized water in a 150 mL flask, and extracted for 2 h at 95C. The extraction procedure was repeated once. The extract was centrifuged at 8000 $\times g$ for 10 min. The combined supernatants were used for the estimation of crude polysaccharides or for further purification by ethanol precipitation when used in the assay of antioxidant activity.

### Ethanol maceration for extraction of triterpenoids

A fine powder (1 g) of *G. lucidum* was extracted with 40 mL of 95% ethanol for 6 h at 30C with gentle shaking in a 150 mL flask. The flask was sealed with aluminum foil during the extraction. The extraction procedure was repeated once. The extraction solution was centrifuged at 8000 $\times g$ for 10 min. The supernatants were combined, and used for measurement of triterpenoids and the assay of antioxidant activity.

### UACE of polysaccharides and triterpenoids

The UACE was performed in an ultrasonic cleaning bath with a 3 L usable capacity (KQ-800KDE, Kun Shan Ultrasonic Instruments Co. Ltd., Jiangsu, China). The ultrasonic frequency was 40 kHz. Each of 1 g of *G. lucidum* fine powder was extracted with an aqueous ethanol in a 150 mL flask, and the other conditions were described elsewhere. The content of the flask was centrifuged at 8000 $\times g$ for 10 min. The polysaccharides and triterpenoids in the resulting supernatant were determined. The yield for polysaccharides or triterpenoids was calculated according to the following equation:

$$\text{Extraction yield } (\%) = \frac{Weight\ of\ polysaccharides\ or\ triterpenoids}{Weight\ of\ G.lucidum\ powder} \times 100 \tag{1}$$

## Desirability function

On the basis of the desirability function [25], the extraction yields of polysaccharides and triterpenoids were compiled into one index (*D* value) according to Eqs (2) and (3):

$$d_i = \begin{cases} 0 & Y_i \leq Y_{\min} \\ \frac{Y_i - Y_{\min}}{Y_{\max} - Y_{\min}} & Y_{\min} < Y_i < Y_{\max} \\ 1 & Y_i \geq Y_{\max} \end{cases} \tag{2}$$

where $Y_i$ is the response value of an *i*-analyzed factor.

$$D = d_1^{w_1} \times d_2^{w_2} \times \ldots \times d_m^{w_m} \tag{3}$$

where $d_i$ represents individual desirability value and $w_i$ is relative importance, which indicates the difference in the importance ascribed to different response variables. In Eq (3), $w_i$ should be in the range of 0−1, and $w_1 + w_2 + \ldots + w_3$ is 1.

According to the preliminary study, the parameters of the desirability function in this study were set as $Y_{min} = 0.1\%$, $Y_{max} = 2.0\%$, importance $w = 0.5$ for polysaccharides, and for triterpenoids, they were as $Y_{min} = 0.05\%$, $Y_{max} = 1.0\%$, and importance $w = 0.5$. The individual desirability value $d_1$ for polysaccharide was calculated based on Eq (4), and the individual desirability value $d_2$ for triterpenoid was calculated according to Eq (5). The overall desirability (*D* value) of the optimization was calculated using the individual desirability values according to Eq (6).

$$d_1 = \begin{cases} 0 & Y_i \leq 0.1\% \\ \frac{Y_i - 0.1\%}{2\% - 0.1\%} & 0.1\% < Y_i < 2\% \\ 1 & Y_i \geq 2\% \end{cases} \tag{4}$$

$$d_2 = \begin{cases} 0 & Y_i \leq 0.05\% \\ \frac{Y_i - 0.05\%}{1\% - 0.05\%} & 0.05\% < Y_i < 2\% \\ 1 & Y_i \geq 1\% \end{cases} \tag{5}$$

$$D = \sqrt{d_1 d_2} \tag{6}$$

## Single-factor experiments

The single-factor experiment was carried out to determine the preliminary range of the extraction variables. The *G. lucidum* fine powder (1 g) was extracted in 150 mL flasks with aqueous ethanol. The optimum extraction temperature was first determined by varying temperature from 40 to 90C under other conditions: 40% ethanol, ultrasonic power 140 W, liquid/solid ratio 40 mL/g, extraction time 60 min, and number of extractions 1. Then, the optimum values of ethanol concentration (20%−80%), ultrasonic power (70−245 W), ratios of liquid to solid (10−60 mL/g), extraction time (30−180 min), and number of extractions (1−4) were determined sequently using determined optimal values in the extraction. Each experiment was conducted in triplicate. The extraction yields of polysaccharides and triterpenoids were determined, and the *D* values were calculated.

**Table 1. Levels and codes of extraction variables used in BBD.**

| Extraction variables | Symbol | | Level[a] | | |
|---|---|---|---|---|---|
| | Coded | Uncoded | -1 | 0 | +1 |
| Ultrasonic power (W) | $X_1$ | $x_1$ | 140 | 175 | 210 |
| Extraction temperature (˚C) | $X_2$ | $x_2$ | 70 | 80 | 90 |
| Liquid/solid ratio (mL/g) | $X_3$ | $x_3$ | 30 | 40 | 50 |
| Extraction time (min) | $X_4$ | $x_4$ | 60 | 90 | 120 |

[a] $X_1 = (x_1-175)/35$; $X_2 = (x_2-80)/10$; $X_3 = (x_3-40)/10$; $X_4 = (x_4-90)/30$

## Box-Behnken Design (BBD) of RSM

After the single factor experiments, four significant variables including $X_1$ (ultrasonic power), $X_2$ (extraction temperature), $X_3$ (liquid/solid ratio), and $X_4$ (extraction time) were selected and further optimized by RSM with BBD [29]. The coded and actual values of the four selected variables are listed in Table 1. A BBD matrix comprising 29 trials was formulated with Design-Expert software (version 8.0), and was shown in Table 2. A total of five replicates (runs 6, 7, 10, 16, and 29) were performed at the center point of the design to allow for estimation of a pure error sum of squares for statistical analysis. To simplify the study, only one extraction number was tested for all experiments in this design. Each experiment was performed in triplicate and average values were shown.

To correlate the relationship between response variables (*D* value) with the four independent variables of the UACE process, the *D* values were fitted to the following second-order quadratic polynomial equation:

$$Y = \beta_0 + \sum_{i=1}^{4} \beta_i X_i + \sum_{i=1}^{4} \beta_{ii} X_i^2 + \sum_{i=1}^{3} \sum_{i<j}^{4} \beta_{ij} X_i X_j \qquad (7)$$

where *Y* is the predicted *D* value, $\beta_0$ is an intercept of the equation, $\beta_i$ is linear coefficient, $\beta_{ii}$ is quadratic coefficient, $\beta_{ij}$ is cross product term coefficient, and $X_i$ and $X_j$ are the coded independent variables. The significance of the polynomial equation was determined by an *F* test. The *P*-value was used as a tool to check the statistical significance of each coefficient in the polynomial equation [30]. The fit of the model was expressed by the coefficient of determination ($R^2$). CV (%) was used as a measure of the accuracy and reliability of the experiments.

## Quantitation of polysaccharides and triterpenoids

The supernatants obtained from the UACE processes were subjected to separation for polysaccharides and triterpenoids before being used for the assay. Appropriate amounts of 95% ethanol were added to the extract supernatants to a final ethanol concentration of 75% and kept for 12 h at 4C for polysaccharide precipitation. After centrifugation, the resulting supernatant was used to determine the concentration of triterpenoids, and the precipitate was washed twice with 95% ethanol, dissolved in distilled water, and used for assay of the polysaccharides. The vanillin-glacial acetic acid method was used for analysis of the triterpenoid with ursolic acid as the standard [24], and expressed as the equivalent amount of ursolic acid. The polysaccharides were measured by the phenol-sulfuric acid method using D-glucose as the standard [31]. The standard curves for estimation of both components are shown in S1 Fig.

## DPPH radical scavenging activity assay

DPPH radical scavenging activity was determined using a method as described previously by Blois [32] with some modifications. Two milliliters of 0.2 mM DPPH in ethanol was added to 2 mL sample at different polysaccharide or triterpenoid concentrations in the range of 0.1−0.5 mg/mL. After vortexing, the mixture was kept for 15 min in the dark at 30C. The absorbance was measured at 517 nm using a UV-Vis spectrophotometer (TU-1810, Beijing Puxi General Instrument Co. Ltd., China). Ethanol was used as the negative control instead of DPPH, as well as the blank. Butylated hydroxytoluene (BHT) was used as a positive control. DPPH scavenging rate was calculated according to the following equation:

$$\text{DPPH scavenging rate } (\%) = \left( 1 - \frac{A_s - A_c}{A} \right) \times 100 \tag{8}$$

where $A_s$ is the absorbance determined for the sample, $A_c$ is the absorbance for the ethanol control, and $A$ is the absorbance for the DPPH solution (2 mL DPPH plus 2 mL ethanol).

**Table 2. BBD matrix and its results for the optimization of UACE process.**

| Run | Coded levels of variables | | | | Extraction yield (%) | | D value |
|-----|------|------|------|------|----------------|---------------|------|
|     | $X_1$ | $X_2$ | $X_3$ | $X_4$ | Polysaccharides | Triterpenoids |      |
| 1   | -1   | -1   | 0    | 0    | 0.45 | 0.35 | 0.24 |
| 2   | 0    | 0    | 1    | 1    | 0.63 | 0.41 | 0.32 |
| 3   | 1    | 1    | 0    | 0    | 0.41 | 0.26 | 0.17 |
| 4   | 0    | 0    | 1    | -1   | 0.27 | 0.33 | 0.16 |
| 5   | 0    | -1   | 1    | 0    | 0.30 | 0.44 | 0.21 |
| 6   | 0    | 0    | 0    | 0    | 0.58 | 0.31 | 0.26 |
| 7   | 0    | 0    | 0    | 0    | 0.72 | 0.34 | 0.31 |
| 8   | -1   | 0    | 0    | -1   | 0.54 | 0.23 | 0.21 |
| 9   | 0    | 1    | 0    | 1    | 0.39 | 0.25 | 0.18 |
| 10  | 0    | 0    | 0    | 0    | 0.68 | 0.32 | 0.29 |
| 11  | 1    | 0    | 0    | 1    | 0.48 | 0.35 | 0.25 |
| 12  | 0    | 0    | -1   | 1    | 0.64 | 0.28 | 0.26 |
| 13  | 0    | 1    | -1   | 0    | 0.26 | 0.11 | 0.07 |
| 14  | 0    | -1   | 0    | -1   | 0.27 | 0.24 | 0.13 |
| 15  | -1   | 0    | 1    | 0    | 0.51 | 0.35 | 0.26 |
| 16  | 0    | 0    | 0    | 0    | 0.66 | 0.31 | 0.28 |
| 17  | 0    | 1    | 0    | -1   | 0.33 | 0.24 | 0.15 |
| 18  | -1   | 0    | -1   | 0    | 0.41 | 0.38 | 0.24 |
| 19  | 1    | 0    | 0    | -1   | 0.50 | 0.36 | 0.26 |
| 20  | 0    | -1   | 0    | 1    | 0.26 | 0.37 | 0.17 |
| 21  | -1   | 1    | 0    | 0    | 0.55 | 0.16 | 0.17 |
| 22  | 0    | 1    | 1    | 0    | 0.29 | 0.30 | 0.16 |
| 23  | 0    | 0    | -1   | -1   | 0.44 | 0.13 | 0.12 |
| 24  | 1    | 0    | 1    | 0    | 0.54 | 0.48 | 0.32 |
| 25  | 1    | -1   | 0    | 0    | 0.22 | 0.29 | 0.13 |
| 26  | 1    | 0    | -1   | 0    | 0.39 | 0.32 | 0.21 |
| 27  | -1   | 0    | 0    | 1    | 0.50 | 0.27 | 0.22 |
| 28  | 0    | -1   | -1   | 0    | 0.14 | 0.28 | 0.07 |
| 29  | 0    | 0    | 0    | 0    | 0.56 | 0.32 | 0.26 |

### Reducing power assay

The reducing capacity of a natural component can employ as a significant indicator of its potential antioxidant activity [33]. A higher absorbance of the reaction mixture at 700 nm shows a higher reducing power [34]. The reducing power of the polysaccharides and triterpenoid-rich extracts was determined following the method described by Ahmadi et al [35], with minor modifications. Briefly, 2 mL sample at varying concentrations (0.1–0.5 mg/mL) was mixed with 2 mL potassium phosphate buffer (0.2 M, pH 6.6), then incubated at 50C for 20 min. The reaction was terminated by adding 2 mL trichloroacetic acid solution (10%, w/v). Then, the reaction solution (2 mL) was mixed with 0.4 mL ferric chloride (0.1%, w/v) and distilled water (2 mL), and the absorbance was measured at 700 nm against a blank after 10 min.

### Ethical statement

This study did not involve any protected species or animals. So any specific permit was not required for the present study.

### Statistical analysis

The experimental data of the BBD were shown as the average of three repetitions, while all other data were represented as the mean ± SD (n = 3). Values of $P < 0.05$ were considered statistically significant. All the graphs were made either with Origin 8.0 (Origin Lab Inc., USA) or Design Expert (version 8.0, Stat-Ease Inc., USA).

## Results

### Extraction yields of polysaccharides and triterpenoids by conventional approaches

The polysaccharides yield of *G. lucidum* obtained by conventional HWE (two numbers of extraction) was 1.52%. For triterpenoids extraction using the conventional maceration method, an extraction yield of 0.59% was obtained using two extraction times.

### Preliminary optimization by one-factor-at-a-time approach

**Effect of extraction temperature on the yields of polysaccharides and triterpenoids.** As presented in Fig 1A, the yield of polysaccharides slowly increased with increasing extraction temperature within the tested range from 40 to 90C. The highest polysaccharide yield was 0.41% at 90C. The triterpenoid yield increased from 0.11% to 0.15% as extraction temperature changed from 40 to 80C, and then decreased above 80C. As to the combined performance, the *D* value increased with the increase of extraction temperature and reached the maximum value of 0.13 when extraction temperature was 80C, and then it started to decline. Thus, an extraction temperature of 70–90C was considered the optimal range in subsequent RSM optimization studies.

**Effect of ethanol concentration on the yields of polysaccharides and triterpenoids.** Fig 1B shows that the polysaccharide yield decreased in a rapid manner with increasing ethanol concentration in the scope of test, with the maximum yield of 0.46% observed at the lowest concentration of 20%. Conversely, the highest triterpenoid yield of 0.34% was obtained at the highest investigated ethanol concentration of 80%. The *D* value increased to 0.15 from 0.04 as ethanol increased from 20% to 50%, then slightly decreased when it increased from 50% to 80%. The optimal ethanol concentration was therefore around 50%. As ethanol concentrations between 40%–60% gave little variation in the *D* value, and the extraction was poor outside of

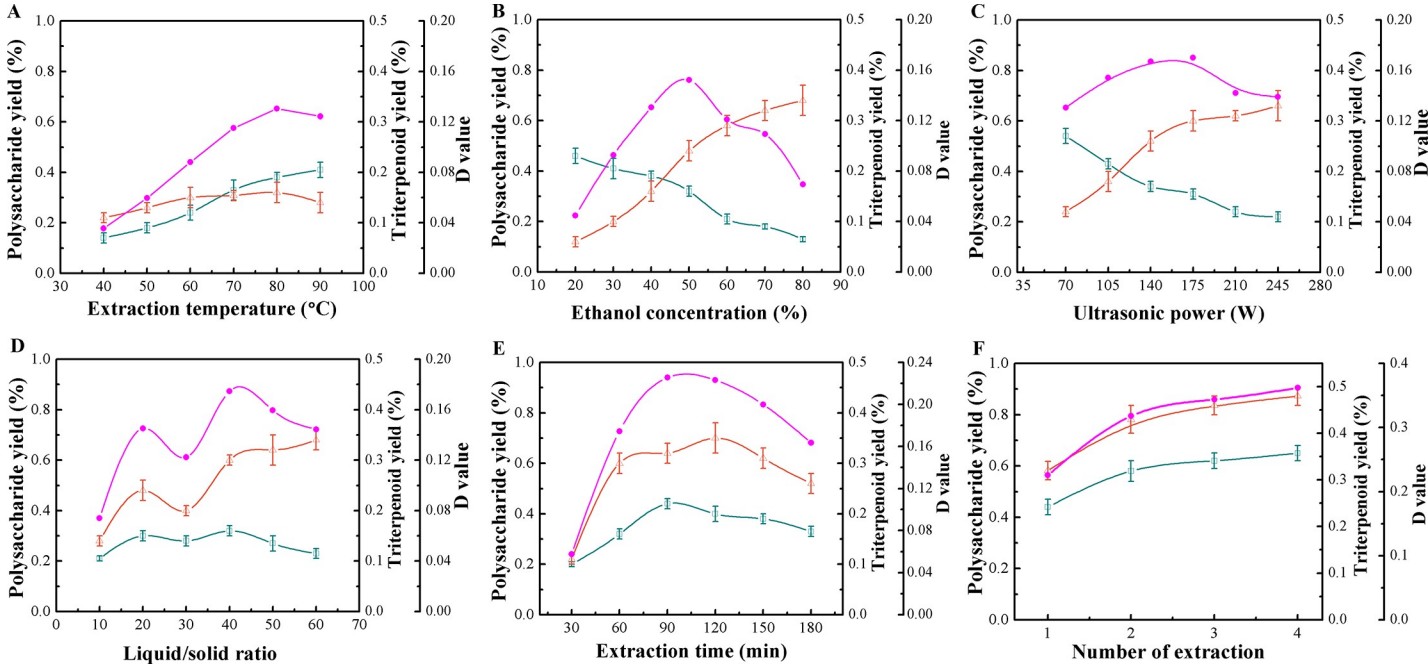

**Fig 1. Effects of different extraction parameters on the yield of polysaccharides and triterpenoids of *G. lucidum*, as well as on the *D* value.** *Open squares*, polysaccharide yield; *open triangles*, triterpenoid yield; and *filled circles*, *D* value.

this range, the ethanol concentration of 50% was selected, and not included in BBD for further optimization.

**Effect of ultrasonic power input on the yields of polysaccharides and triterpenoids.** Fig 1C shows that the optimal ultrasonic power inputs for extraction of polysaccharides versus triterpenoids were opposite: maximum yield for polysaccharides was 0.54% at the lowest studied ultrasonic power (70 W), whereas the highest yield of triterpenoids, reaching 0.33%, was obtained when ultrasonic power applied was the highest at 245 W. As the ultrasonic power increased, the *D* value rose and reached a peak value of 0.17 at 175 W. An ultrasonic power of 175 W was therefore desirable in the extraction.

**Effect of ratios of liquid to solid on the yields of polysaccharides and triterpenoids.** As presented in Fig 1D, the yield of polysaccharides increased from 0.21% to 0.3% as the liquid/solid ratio rose from 10 to 20 mL/g but decreased as the ratio further increased to 30 mL/g, while a yield increase from 0.28% to 0.32% was observed when the liquid/solid ratio ascended from 30 to 40 mg/L. Afterward, the polysaccharide yield decreased gradually. The maximum yield of polysaccharides (0.32%) was obtained at a liquid/solid ratio of 40 mL/g. The yield of triterpenoids increased with the increasing liquid/solid ratio within the tested range except when the liquid/solid ratio was 20 mL/g, with a maximum yield of 0.34% at a ratio of 60 mL/g. As to the combined extraction efficacy, the maximum *D* value (0.17) was shown to be at a liquid/solid ratio of 40 mL/g, hence, this ratio was considered as the center point in the subsequent BBD.

**Effect of extraction time on the yields of polysaccharides and triterpenoids.** From Fig 1E, the polysaccharide yield increased with extraction time, until it reached the maximum value (0.44%) when extraction time was 90 min. With further extended time, the yield decreased. Triterpenoid yield increased from 0.11% to 0.35% as extraction time was increased from 30 to 120 min, and then it started to decrease. Regarding the combined performance, the

*D* value increased first and reached a maximum value (0.23) when the extraction time was 90 min, and then decreased with time. An extraction time of 90 min was therefore considered desirable in subsequent studies.

**Effect of number of extractions on the yields of polysaccharides and triterpenoids.** As can be seen from Fig 1F, the yields of both polysaccharides (0.44%) and triterpenoids (0.32%) were high after the first extraction, and only small amounts of both components were extracted from the residue after an additional extraction. After extraction number two, almost no additional polysaccharides and triterpenoids could be extracted. The changing trend of *D* values against extraction number was the same for extraction yields of both polysaccharides and triterpenoids.

## Optimization of the UACE process by RSM

**Model fitting and statistical analysis.** The matrix of BBD and experimental results are shown in Table 2. By applying multiple regression analysis on the *D* values, the following second-order quadratic polynomial equation in coded values was obtained:

$$Y = 0.28 + 0.000825X_1 - 0.003217X_2 + 0.039X_3 + 0.03X_4 + 0.031X_1X_2 + 0.023X_1X_3 - $$
$$0.005175X_1X_4 - 0.012X_2X_3 - 0.00245X_2X_4 + 0.0058X_3X_4 - 0.002946X_1^2 - 0.11X_2^2 - \quad (9)$$
$$0.036X_3^2 - 0.031X_4^2$$

where *Y* is the predicted *D* value; $X_1$, $X_2$, $X_3$, and $X_4$ are the coded independent variables for ultrasonic power input, extraction temperature, ratio of liquid/solid, and extraction time, respectively.

The data of statistical analysis on the fitted response surface quadratic model are listed in Table 3. The high model *F*-value (4.46) and a low probability value (0.0042) indicated that the model was statistically significant. The low lack-of-fit *F*-value (4.63) and its corresponding high *P*-value (0.0765) indicated that the lack of fit was insignificant relative to the pure error and the model was reliable. The determination coefficient ($R^2$) of the regression model was 0.817, indicating that more than 81% of total variability in the response could be explained using this model. The signal-to-noise ratio was as high as 6.952, and therefore the model was considered fit and could be used to navigate the design space. *P*-values of linear variables $X_3$ and $X_4$, and quadratic variable $X_2^2$ were lower than 0.05, thus, they were significant model terms, and had significant influences on the extraction.

**Response surface analysis.** The graphical representation of response surface for Eq (9) is shown in Fig 2. The effect of ultrasonic power and extraction temperature on the *D* value and their interactions are depicted in Fig 2A. The *D* value increased as extraction temperature increased from 70 to approximately 80C, and then it started to decrease, but there was only a little change when varying ultrasonic power in the investigated range. The projection of the response surface indicated that there was no significant interaction between these two extraction variables. From Fig 2B, a high *D* value could be obtained by applying high levels of both ultrasonic power input and ratio of liquid to solid. When the liquid/solid ratio was low but ultrasonic power input was high, low *D* values were readily observed, implying the poor performance of the extraction.

From Fig 2C, a longer ultrasonic time helped to extract polysaccharides and triterpenoids, whereas the change in ultrasonic power input had little effect on the *D* value. The surface plot presents a flat plain, indicating that they had little interaction. Fig 2D shows that a small change in either the liquid/solid ratio or extraction temperature would lead to a considerable variation in the *D* value, and higher *D* values would be attained when a high liquid/solid ratio along with moderate extraction temperature were simultaneously applied in the extraction.

**Table 3. Analysis of variance for the fitted response surface quadratic model.**

| Source | Sum of squares | DF | Mean square | F-value | P-value |
|---|---|---|---|---|---|
| Model | 0.11 | 14 | 8.105E − 03 | 4.46 | 0.0042 |
| $X_1$: Ultrasonic power | 8.167E − 06 | 1 | 8.167E − 06 | 4.498E − 03 | 0.9475 |
| $X_2$: Extraction temperature | 1.242E − 04 | 1 | 1.242E − 04 | 0.068 | 0.7975 |
| $X_3$: Liquid/solid ratio | 0.018 | 1 | 0.018 | 9.95 | 0.0070 |
| $X_4$: Ultrasonic time | 0.011 | 1 | 0.011 | 5.96 | 0.0285 |
| $X_1 \times X_2$ | 3.788E − 03 | 1 | 3.788E − 03 | 2.09 | 0.1706 |
| $X_1 \times X_3$ | 2.186E − 03 | 1 | 2.186E − 03 | 1.20 | 0.2911 |
| $X_1 \times X_4$ | 1.071E − 04 | 1 | 1.071E − 04 | 0.059 | 0.8116 |
| $X_2 \times X_3$ | 5.546E − 04 | 1 | 5.546E − 04 | 0.31 | 0.5892 |
| $X_2 \times X_4$ | 2.401E − 05 | 1 | 2.401E − 05 | 0.013 | 0.9101 |
| $X_3 \times X_4$ | 1.346E − 04 | 1 | 1.346E − 04 | 0.074 | 0.7894 |
| $X_1^2$ | 5.629E − 05 | 1 | 5.629E − 05 | 0.031 | 0.8628 |
| $X_2^2$ | 0.072 | 1 | 0.072 | 39.88 | < 0.0001 |
| $X_3^2$ | 8.317E − 03 | 1 | 8.317E − 03 | 4.58 | 0.0504 |
| $X_4^2$ | 6.343E − 03 | 1 | 6.343E − 03 | 3.49 | 0.0827 |
| Residual | 0.025 | 14 | 1.816E − 03 | | |
| Lack of fit | 0.023 | 10 | 2.340E − 03 | 4.63 | 0.0765 |
| Pure error | 2.023E − 03 | 4 | 5.057E − 04 | | |
| Cor total | 0.14 | 28 | | | |
| $R^2$ = 0.8170 | | | | | |
| Adj $R^2$ = 0.6339 | | | | | |
| Adeq precision = 6.952 | | | | | |
| CV(%) = 20.19 | | | | | |

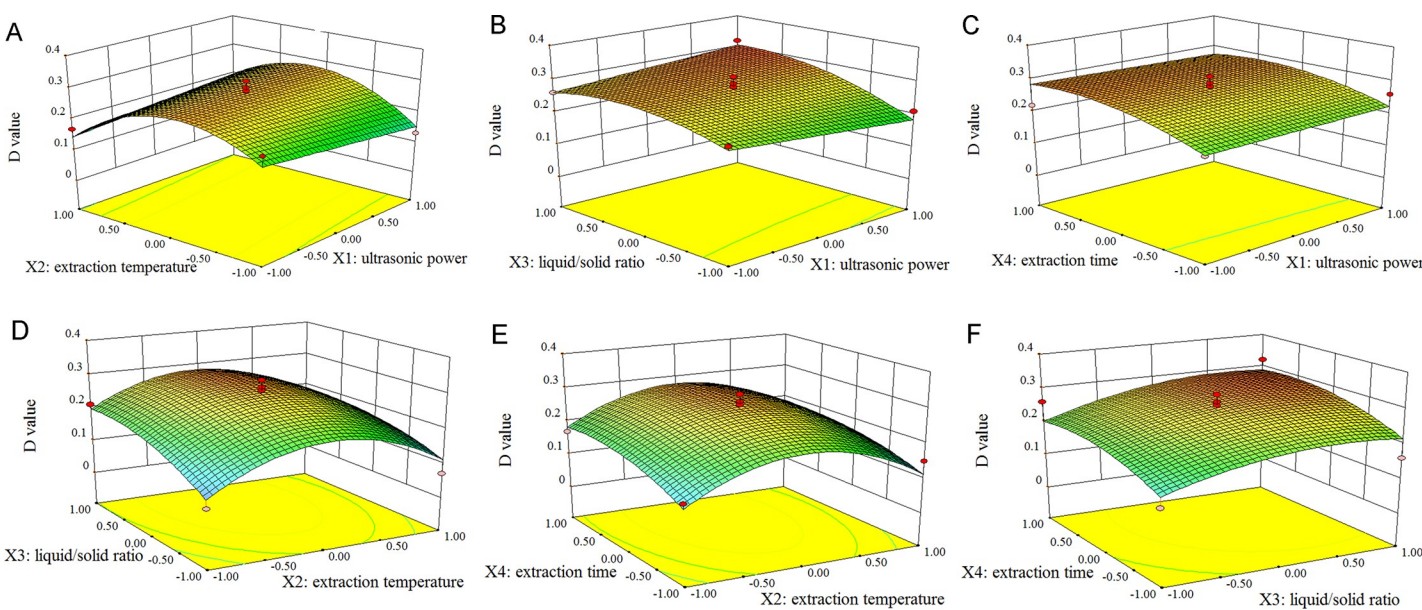

**Fig 2. Response surface plots showing the effects of extraction parameters on the *D* values in the extraction of polysaccharides and triterpenoids from *G. lucidum*.** A, effect of ultrasonic power and extraction temperature; B, effect of ultrasonic power and liquid/solid ratio; C, effect of ultrasonic power and extraction time; D, effect of extraction temperature and liquid/solid ratio; E, effect of extraction temperature and extraction time; and F, effect of liquid/solid ratio and extraction time.

As is evident from Fig 2E, the *D* value increased first and then declined with time. Extraction temperature had an effect on the *D* value similar to extraction time. The maximum *D* value was attained at around intermediate levels (extraction temperature of 80C and 90 min extraction time) for both variables. As observed in Fig 2F, the *D* value first increased with extraction time, then decreased when extraction time reached a high level (around 105 min). At all investigated extraction times, the *D* value increased first, then decreased when the liquid/solid ratio was greater than 50 mL/g.

**Optimization of extraction parameters of UACE.** Through the numerical optimization technique of the Design-Expert software, the optimal conditions of UACE for the extraction of *G. lucidum* polysaccharides and triterpenoids, with the *D* value as the response variable, were determined to be the following: ultrasonic power 210 W, extraction temperature 80.77C, ratio of liquid/solid 49.85 mL/g, and extraction time 103.56 min. In consideration of actual experimental conditions, optimal extraction conditions were modified as ultrasonic power 210 W, extraction temperature 80C, ratio of liquid/solid 50 mL/g, and extraction time 100 min.

**Validation of predictive model.** The adequacy and reliability of the regression model was verified using three independent experiments in duplicate under the modified conditions. The experimental yield of polysaccharides was 0.63% and the triterpenoid yield reached 0.38%. The *D* value was calculated to be 0.31, which was generally consistent with the predicted value of 0.33 given by the model. The verification experiment confirmed the precision and reliability of the model.

## Comparison of DPPH radical scavenging activity of the polysaccharides and triterpenoid-rich extracts between optimized UACE and conventional processes

Fig 3A shows that the DPPH scavenging activity of polysaccharides obtained by optimized UACE and HWE showed a dosage-dependent manner, and the activity of the polysaccharides obtained by optimal UACE was higher than that of HWE in the tested concentration range. At

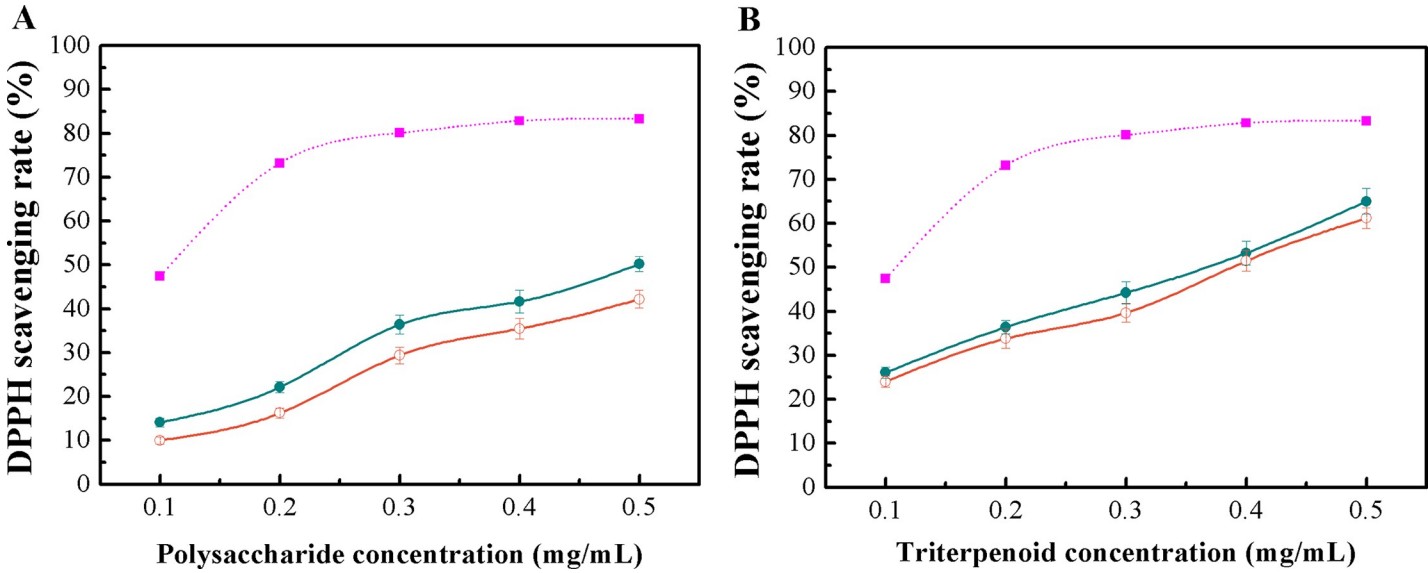

**Fig 3.** DPPH radical scavenging activity of the polysaccharides (A) and triterpenoid-rich extracts (B) from optimal UACE and traditional extraction methods. *Filled circles*, polysaccharides or triterpenoid-rich extracts from optimal UACE; *open circles*, polysaccharides from HWE or triterpenoid-rich extracts from ethanol maceration, and *filled squares*, BHT.

0.2 mg/mL, the scavenging rate of the polysaccharides from the optimized UACE was 22.16%, higher than that (16.24%) of the HWE polysaccharides. Regardless of extraction methods, the DPPH scavenging rates of polysaccharides obtained were considerably lower than that of BHT at the same concentration. The results in Fig 3B indicated that there were no significant differences in the DDPH radical scavenging activity between the triterpenoid-rich extracts obtained by the optimized UACE process and the maceration method at all tested concentrations, and the activity of the two triterpenoid-rich extracts was strongly dependent on their concentration.

## Comparison of reducing power of the polysaccharides and triterpenoid-rich extracts between optimized UACE and conventional processes

As shown in Fig 4A, the reducing power of polysaccharides obtained by optimal UACE and HWE methods exhibited a dosage-dependent pattern, and the polysaccharides obtained by optimal UACE showed a stronger reducing power than those extracted by the HWE method in the tested concentration range. At 0.2 mg/mL, the absorbance at 700 nm was 0.34 for the polysaccharides from optimal UACE, while it was only 0.24 for the HWE polysaccharides. The reducing power for both extracted polysaccharides was lower than that of BHT at the same concentration. Fig 4B shows that the reducing power of the triterpenoid-rich extracts obtained by the optimized UACE process and the ethanol maceration method was strongly dependent on their concentration, and there were no significant differences between the two extracts at all tested concentrations. The reducing power for both triterpenoid-rich extracts was considerably lower than that of the positive standard BHT.

## Discussion

The efficient extraction of polysaccharides and triterpenoids from *G. lucidum* is a prerequisite for their further research and industrial application. Hence, studying the extraction of this medicinal mushroom for these two active ingredients is of particular importance. In the present investigation, the environmentally friendly ultrasonic-assisted extraction technology was

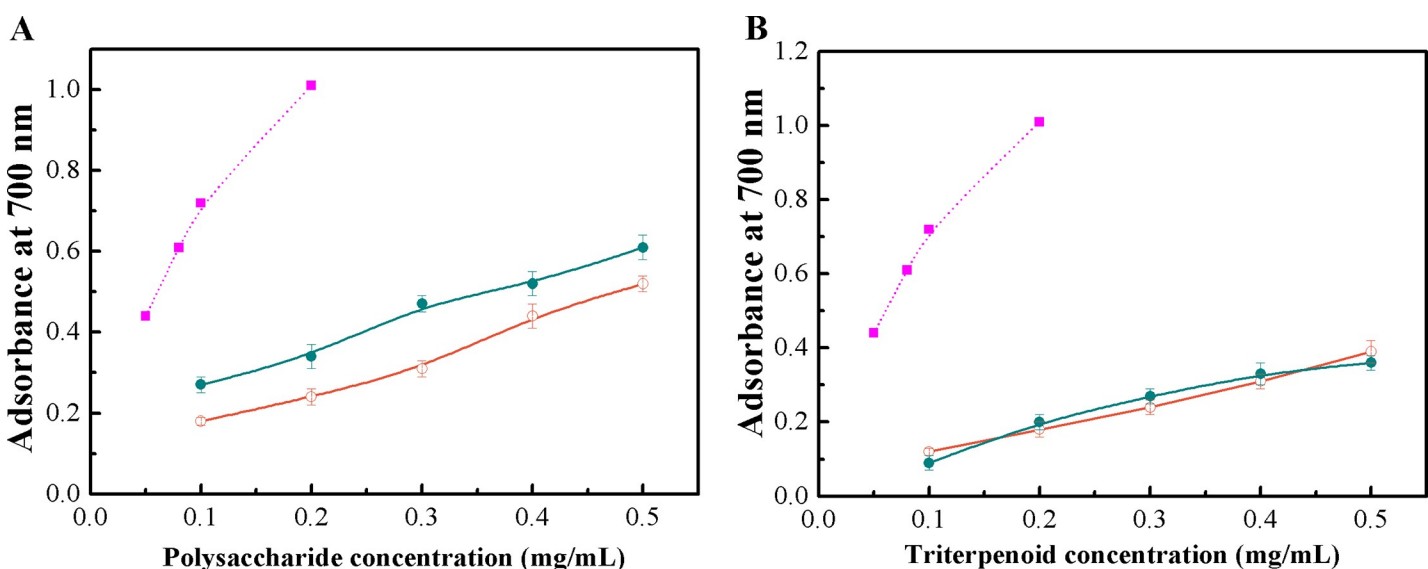

**Fig 4.** Reducing power of the polysaccharides (A) and triterpenoid-rich extracts (B) from optimal UACE and conventional methods. *Filled circles*, polysaccharides or triterpenoid-rich extracts from optimal UACE; *open circles*, polysaccharides from HWE or triterpenoid-rich extracts from ethanol maceration; and *filled squares*, BHT.

employed for the simultaneous extraction of polysaccharides and triterpenoids from *G. lucidum* in a simple process (S2 Fig), with a resulting satisfactory performance. The success of synchronously extracting these two contrasting components (in terms of water solubility) by the developed UACE process could be attributed to the synergistic action of an appropriate extraction solvent, a relatively high extraction temperature, and acoustic cavitations generated by ultrasound. In this process, the polysaccharides and triterpenoids might not dissolve well in the extraction solvent applied, while a relatively high temperature increases their solubility, and finally ultrasound accelerates the extraction. To the best of our knowledge, the present study is the first report on the simultaneous extraction of polysaccharides and triterpenoids from *G. lucidum* in a simple extraction process.

In the present study, the extraction yields of *G. lucidum* polysaccharides and triterpenoids using the optimized UACE process were 0.63% and 0.38%, respectively. These two yields were lower than those obtained with conventional processes (polysaccharide yield of 1.52% by HWE and triterpenoid yield of 0.59% by ethanol maceration). The obtained polysaccharide yield was also lower than that observed in the ultrasonic study done by Ma et al [8], where a yield of 2.87% was reported, and that of Kan et al [36], which showed 2.44% of polysaccharide yield. The low polysaccharide yield in this study may be largely due to its low solubility in aqueous ethanol, despite the use of ultrasound in the extraction. Regarding triterpenoids, the extraction yield was lower than that of the reported yield (0.97%) using microwave irradiation [7]. A reasonable explanation is that the solubility of some low polarity or non-polar triterpenoid compounds in this mushroom may be low in the extracting solvent.

Extraction processes influencing the bioactivity of extracts have been reported widely [37, 38]. For fungal polysaccharides, a large number of factors, including their chemical components, molecular mass, structure, and conformation, can remarkably affect their antioxidant activities [8, 39]. In the present study, the polysaccharides obtained by optimized UACE showed a higher antioxidant activity than that of polysaccharides obtained by the HWE method. Ultrasonic treatment in the extraction leading to the increase in antioxidant activity of the polysaccharides of longan fruit pericarp was also demonstrated, and the underlying mechanism has been attributed to the degradation of high molecular weight polysaccharides and further changes in chemical structure caused by acoustic cavitation effects [40]. The antioxidant activity of the polysaccharides from common mullein (*Verbascum thapsus* L.) flowers was also shown to be enhanced by ultrasound in the extraction [41]. We therefore postulated that the degradation of polysaccharides by the action of ultrasound and a higher portion of low-molecular-weight polysaccharides extracted from the matrice may be responsible for the higher antioxidant activity. The difference in the polysaccharide purity between the two extracted polysaccharides may also be a factor.

The triterpenoids of *G. lucidum* are related to its many important biological activities. In the present study, based on the DPPH radical scavenging activity and evaluation of reducing power, the antioxidant activity of triterpenoid-rich extracts of *G. lucidum* obtained under optimized UACE process did not alter by ultrasonic treatment in the process when compared with those obtained using the conventional ethanol maceration method. This may be because the triterpenoids have a relatively simple and stable chemical structure compared with the high-molecular-weight polysaccharides. Therefore, the ultrasonic conditions applied in this work did not lead to a significant detrimental effect on their chemical properties and therefore bioactivity. However, further studies would be needed to investigate the change in patterns of extracted triterpenoids of *G. lucidum* during the extraction by ultrasound.

## Conclusion

In the present study, the simultaneous and efficient extraction of polysaccharides and triterpenoids from the medicinal fungus *G. lucidum* was achieved with ultrasonic extraction technology. The optimized extraction conditions by RSM with a desirability function were determined as: ultrasonic power of 210 W, extraction temperature of 80C, liquid/solid ratio of 50 mL/g, and extraction time of 100 min. On the basis of the scavenging capacity of DPPH and evaluation of reducing power, the antioxidant activity of the extracted polysaccharides by ultrasonic action in optimal UACE process was significantly enhanced, while the triterpenoid-rich extracts demonstrated unchanged activity. The present study is the first report on the application of ultrasound for the simultaneous extraction of polysaccharides and triterpenoids from *G. lucidum* in a convenient extraction process.

## Supporting information

**S1 Fig.** Standard curves for estimation of polysaccharides (A) and triterpenoids (B) used in this work. A, with d-glucose as the standard, and B, with ursolic acid as the standard.
(TIF)

**S2 Fig. Flow chart of the UACE process.**
(TIF)

## Author Contributions

**Conceptualization:** Shengrong Liu.

**Data curation:** Shizhong Zheng, Shengrong Liu.

**Formal analysis:** Shengrong Liu.

**Funding acquisition:** Shengrong Liu.

**Investigation:** Shizhong Zheng, Weirui Zhang.

**Methodology:** Shengrong Liu.

**Project administration:** Shengrong Liu.

**Supervision:** Shengrong Liu.

**Validation:** Shizhong Zheng, Weirui Zhang.

**Writing – original draft:** Shizhong Zheng, Shengrong Liu.

**Writing – review & editing:** Shizhong Zheng, Weirui Zhang, Shengrong Liu.

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
