## [Decision Letter · Decision Letter 0]

29 Sep 2020

PONE-D-20-24854

Desirability function approach for optimization of ultrasound-assisted extraction of polysaccharides and triterpenoids from Ganoderma lucidum and evaluation of their in vitro antioxidant capacities

PLOS ONE

Dear Dr. Liu,

Thank you for submitting your manuscript to PLOS ONE. After careful consideration, we feel that it has merit but does not fully meet PLOS ONE’s publication criteria as it currently stands. Therefore, we invite you to submit a revised version of the manuscript that addresses the points raised during the review process.

It is rather out of the point to determine the antioxidant activity of a mix of compounds, particularly using only DPPH assay as noted by Reviewer #3. I can suggest the authors to isolate dominant polysaccharides and triterpenes and to further test their antioxidant activities employing multiple common assays in order to match the theme of the main manuscript title. One of the possibilities can be to shift the manuscript theme to the extraction methodology solely (as proposed by Reviewer #3), which, in the other hand, would decrease the manuscript informativeness. The Discussion section must deepened as per reviewers' suggestions. Language should be considerably revised in order to increase readability and overall presentation clarity.

We look forward to receiving your revised manuscript.

Kind regards,

Branislav T. Šiler, Ph.D.

Academic Editor

PLOS ONE

Journal Requirements:

Reviewers' comments:

Reviewer's Responses to Questions

**Comments to the Author**

1. Is the manuscript technically sound, and do the data support the conclusions?

Reviewer #1: Partly

Reviewer #2: Yes

Reviewer #3: Partly

2. Has the statistical analysis been performed appropriately and rigorously? 

Reviewer #1: Yes

Reviewer #2: Yes

Reviewer #3: Yes

3. Have the authors made all data underlying the findings in their manuscript fully available?

Reviewer #1: Yes

Reviewer #2: Yes

Reviewer #3: No

4. Is the manuscript presented in an intelligible fashion and written in standard English?

Reviewer #1: No

Reviewer #2: Yes

Reviewer #3: Yes

5. Review Comments to the Author

Reviewer #1: The manuscript describes the antioxidant capacity of polysaccharides and triterpenoids obtained from Ganoderma lucidum, using ultrasonic-assisted co-extraction (UACE) and response surface methodology (RSM). The paper is interesting but need major revision.

The MS title is not appropriate and need revision.

The abstract is generalized. Incorporate your significant data into the abstract.

In introduction, the extraction procedures are focused rather than the importance of Ganoderma constituents and their ability to cure various ailments.

In materials and method section, references are missing in subsections. Add pertinent references to the methods adopted.

In results section, the authors have added methods too, which needs to be shifted to the relevant section. For instance line 265-268.

The discussion is very shallow and needs major input.

The linguistic quality of the MS is very poor (for instance line 237). Edit the MS drastically for grammatical mistakes and typos.

I would suggest the authors to make a graphical model incorporating all information presented in the paper.

The authors may add the recent relevant reference, Ryua DH, JY Cho, NB Sadiq, JC Kim, BY Lee, M Hamayun et al. (2020) Optimization of antioxidant, anti-diabetic, and anti-inflammatory activities and ganoderic acid content of differentially dried Ganoderma lucidum using response surface methodology. Food Chemistry, 335: 127645

Reviewer #2: The simultaneous recovery of polysaccharides and triterpenoids were reported by the authors using ultrasound-assisted extraction. The authors first performed one factor at a time experiment in order to determine the domain of the experimental design and optimised the influence of four significant variables (ultrasonic power, temperature, liquid/solid ratio, and extraction time). The introduction of the manuscript is well prepared and follows the theme of the article. A good methodology was established, the authors provide a clear rationale for each experiment conducted and the results are discussed thoroughly. Discussions are very well reported, and the authors focused on comparing their finding with previous reports. The figures are self-explanatory, clear, and easy to understand. However, I provided a few suggestions and recommend the acceptance of this paper in PLOSONE.

1. Please rephrase line 30-33 in the abstract for more clarity

2. Line 45: Include the classificator of Ganoderma lucidum mushroom for the first time (Karst.)

3. Line 37-38: "the triterpenoids showed antioxidant activities similar to those isolated using the ethanol maceration" This sentence is not correct because the authors performed only extraction. Hence, it should be corrected as "the triterpenoid rich extract showed antioxidant activities similar to those obtained using ethanol maceration.

4. Rephrase line 76

5. Line 208-210: The quantification of triterpenes should be better explained. Dis the authors used a standard (Ursolic acid for example). If yes, then the results should be expressed in the equivalent of the standard

6. In its current state, the manuscript has some grammatical errors and instances of badly constructed sentences. Please check the manuscript and refine the language carefully.

Reviewer #3: This study describes a methodology for simultaneously extracting polysaccharides and triterpenes.

However, the goals of bioactivity differ greatly between polysaccharides and triterpenes. For example, there are many types of triterpenes, and each triterpene has a unique bioactivity. Isn't each activity weakened by simultaneous extraction?

In this study, the author describes the usefulness of simultaneous extraction, but I think it is not enough to confirm with DPPH alone.If you have a reason, please state why you expressed the bioactivity only with the DPPH result. In some cases, I suggest removing the DPPH results and changing to a paper with only extraction methods.

P3L62: Please add a description of GA-MT and GA-T

P5 Materials and Methods: Particle size after sample grinding should be specified

P6L119-L147: The description of the extraction method is redundant. Please correct it to a concise description.

Figures: I can't consider the data because the figure doesn't have a legend.

6. PLOS authors have the option to publish the peer review history of their article (what does this mean?). If published, this will include your full peer review and any attached files.

Reviewer #1: No

Reviewer #2: No

Reviewer #3: No

---

## [Author Response · Author response to Decision Letter 0]

25 Nov 2020

Responses to the reviewer’s comments

Manuscript number: PONE-D-20-24854

Title: Optimization of ultrasonic-assisted extraction of polysaccharides and triterpenoids from the medicinal mushroom Ganoderma lucidum and evaluation of their in vitro antioxidant capacities

Dear Editors,

We appreciate your comments and suggestions on our manuscript submitted previously. The manuscript has been carefully revised according to the suggestions and recommendation by the reviewers, and now all problems have been addressed in revised manuscript. Revised portion are marked in red in the paper. Point-by-point responses to these comments are showed below in this letter.

Sincerely yours,

Shi-Zhong Zheng, Wei-Rui Zhang, Sheng-Rong Liu

Reviewer 1#: Some doubts, suggestions and recommendation

Comment 1: The manuscript describes the antioxidant capacity of polysaccharides and triterpenoids obtained from Ganoderma lucidum, using ultrasonic-assisted co-extraction (UACE) and response surface methodology (RSM). The paper is interesting but need major revision.

Response: Thanks. We have carefully and extensively revised the manuscript.

Comment 2: The manuscript title is not appropriate and need revision.

Response: Accept. It has been revised.

Comment 3: The abstract is generalized. Incorporate your significant data into the abstract.

Response: Thanks. The abstract has been carefully revised as suggested, and the quality was greatly improved. See in the revision.

Comment 4: In introduction, the extraction procedures are focused rather than the importance of Ganoderma constituents and their ability to cure various ailments.

Response: Thanks. Some abundant description has been deleted, and the focus has been paid to the extraction techniques and optimization. See in the revision.

Comment 5: In materials and method section, references are missing in subsections. Add pertinent references to the methods adopted.

Response: Accept. Several related references have been cited in the revised manuscript. 

Comment 6: In results section, the authors have added methods too, which needs to be shifted to the relevant section. For instance line 265-268.

Response: Accept. The extraction conditions were deleted in Results section, and described in Materials and methods section. See in the revision.

Comment 7: The discussion is very shallow and needs major input.

Response: Thanks. We have carefully revised this section, and more in-depth explanation has been provided. See in the revision.

Comment 8: The linguistic quality of the MS is very poor (for instance line 237). Edit the manuscript drastically for grammatical mistakes and typos.

Response: Thanks. We have carefully checked the whole manuscript.

Comment 9: I would suggest the authors to make a graphical model incorporating all information presented in the paper.

Response: Accept. A graphical model was made and provided. 

Comment 10: The authors may add the recent relevant reference, Ryua DH, JY Cho, NB Sadiq, JC Kim, BY Lee, M Hamayun et al. (2020) Optimization of antioxidant, anti-diabetic, and anti-inflammatory activities and ganoderic acid content of differentially dried Ganoderma lucidum using response surface methodology. Food Chemistry, 335: 127645

Response: Accept. It has been cited in the manuscript. See in the revision.

Reviewer 2#: Some doubts, suggestions and recommendation

Comment 1: Please rephrase line 30-33 in the abstract for more clarity.

Response: Accept. We has revised it for clarity.

Comment 2: Line 45: Include the classificatory of Ganoderma lucidum mushroom for the first time (Karst.)

Response: Accept. It has been described. See in the revision.

Comment 3: Line 37-38: “the triterpenoids showed antioxidant activities similar to those isolated using the ethanol maceration" This sentence is not correct because the authors performed only extraction. Hence, it should be corrected as” the triterpenoid-rich extract showed antioxidant activities similar to those obtained using ethanol maceration.

Response: Thanks. It has been revised as suggested.

Comment 4: Rephrase line 76: “For extraction of triterpenoids of G. lucidum, maceration using organic extracting solvents such as ethanol is often used”.

Response: Accept. It has been rephrased. See in the revision.

Comment 5: Line 208-210: The quantification of triterpenes should be better explained. Did the authors use a standard (ursolic acid for example). If yes, then the results should be expressed in the equivalent of the standard.

Response: Accept. We used D-glucose as the standard for polysaccharides measurement, and ursolic acid as the standard for the measurement of triterpenoids. The results were expressed in the equivalent of the standard. There information have been described in the manuscript.

Comment 6: In its current state, the manuscript has some grammatical errors and instances of badly constructed sentences. Please check the manuscript and refine the language carefully.

Response: Thanks. We have carefully checked the manuscript.

Reviewer 3#: Some doubts, suggestions and recommendation

Comment 1: This study describes a methodology for simultaneously extracting polysaccharides and triterpenes. However, the goals of bioactivity differ greatly between polysaccharides and triterpenes. For example, there are many types of triterpenes, and each triterpene has a unique bioactivity. Isn’t each activity weakened by simultaneous extraction?

Response: Thanks. We are very sorry that the separation of individual triterpenoids was not carried out, and the activity of each triterpene was not investigated in this study.

Comment 2: In this study, the author describes the usefulness of simultaneous extraction, but I think it is not enough to confirm with DPPH alone. If you have a reason, please state why you expressed the bioactivity only with the DPPH result.

Response: Thanks. For a systematic comparison, we have performed additional experiments on the reducing power of the extracts, and the results have been provided in the manuscript. See in the revision.

Comment 3: In some cases, I suggest removing the DPPH results and changing to a paper with only extraction methods.

Response: Thanks. We provided additional results on the antioxidant assay, and the results on the antioxidant assay were present in the manuscript. See in the revision.

Comment 4: Page 3, Line 62: Please add a description of GA-MT and GA-T

Response: Accept. It has been added.

Comment 5: Page 5 Materials and Methods: Particle size after sample grinding should be specified.

Response: Accept. The particle size was specified in the revision.

Comment 6: Page 6, L119-L147: The description of the extraction method is redundant. Please correct it to a concise description.

Response: Accept. We have shortened it. See in the revision.

Comment 7: Figures: I can’t consider the data because the figure doesn’t have a legend.

Response: Accept. Legends have been provided. See in the revision.

List of the changes in revision

Title

1. Line 1-3 of page 1, “Desirability function approach for optimization of ultrasound-assisted extraction of polysaccharides and triterpenoids from Ganoderma lucidum and evaluation of their in vitro antioxidant capacities” was revised as “Optimization of ultrasonic-assisted extraction of polysaccharides and triterpenoids from the medicinal mushroom Ganoderma lucidum and evaluation of their in vitro antioxidant capacities”.

Abstract

1. Line 24-25 of page 2, “Ganoderma lucidum is a medicinal mushroom that is rich in biologically active substances” was revised as “Ganoderma lucidum (Fr.) Krast, commonly known as “Lingzhi” in Chinese, is a medicinal mushroom that is rich in biologically active substances”.

2. Line 25 of page 2, “and widely used as a traditional Chinese herbal medicine in Asian countries” was deleted.

3. Line 26-27 of page 2, “Polysaccharides and triterpenoids are the two major bioactive ingredients of this mushroom” was changed as “Polysaccharides and triterpenoids are the two major components responsible for the bioactivity of this fungus”.

4. Line 28 of page 2, “of” was revised as “from”.

5. Line 29 of page 2, “with the equal importance for the two components” was inserted.

6. Line 29 of page 2, “single factor tests” was revised as “Following single factor experiments”.

7. Line 30-31 of page 2, “by RSM with the associated D value as the response variable” was deleted.

8. Line 33 of page 2, “With” was changed as “Under”.

9. Line 35 of page 2, “and evaluation of reducing power” was inserted.

10. Line 36 of page 2, “crude” was inserted.

11. Line 37 of page 2, “crude” was inserted.

12. Line 37-38 of page 2, “while the triterpenoids showed” was revised as “whereas the triterpenoid-rich extracts”.

13. Line38-39 of page 2, “using the ethanol maceration” was changed as “using the ethanol maceration method”.

14. Line 39 of page 2, “The present study is the first report on the simultaneous extraction of polysaccharides and triterpenoids from G. lucidum” was inserted.

Keywords

1. Line 42-43 of page 2, “ultrasonic extraction” was inserted.

Introduction

1. Line 45 of page 3, “Ganoderma lucidum is the most famous medicinal mushroom in the world” was revised as “Ganoderma lucidum (Fr.) Krast belonging to the family of Polyporaceae, commonly known as “Lingzhi” in Chinese and “Reishi” in Japanese, is one of the most famous medicinal mushrooms”.

2. Line 47 of page 3, “in oriental countries” was revised as “in many oriental countries”.

3. Line 48-51 of page 3, “Nowadays, this mushroom is widely used to prevent and treat many challenging human diseases, such as hepatitis, hyperglycemia, high blood pressure, and cancers, and as a nutraceutical to improve health, and promote spiritual growth for human in many countries” was deleted.

4. Line 51-53 of page 3, “Owing to its medicinal properties and perceived health-promoting effects, there has been a growing worldwide interest in utilizing G. lucidum in the biomedical industry in recent years” was revised as “In recent years, there has been a growing worldwide interest in utilizing G. lucidum in the biomedical industry due to its pharmacological properties”.

5. Line 53 of page 3, “many G. lucidum products” was revised as “a variety of G. lucidum products”.

6. Line 54 of page 3, “in many parts of the world” was inserted.

7. Line 55 of page 3, “G. lucidum contains” was revised as “G. lucidum in the form of fruiting bodies, mycelium, and spores contains”.

8. Line 57-59 of page 3, “the two key bioactive components namely polysaccharides and triterpenoids are receiving special attention from researchers in the field of biomedicine” was revised as “the polysaccharides and triterpenoids are receiving special attention from researchers in the field of biomedicine”.

9. Line 59 of page 3, “Several kinds of polysaccharides” was revised as “Several polysaccharides”.

10. Line 60 of page 3, “are recognized” was revised as “generally recognized”.

11. Line 60 of page 3, “such as antitumor, antioxidant and immunomodulatory effects” was inserted.

12. Line 61-63 of page 3, “Regarding various chemically diverse triterpenoids, ganoderic acids such as GA-Me and GA-T have been proven to have potent pharmacological activity” was revised as “The triterpenoids from this fungus exhibit structural diversity, and many of them such as ganoderic acids Me and T (a kind of highly oxygenated lanostane-type triterpenoids) have been proven to have potent antitumor activity”.

13. Line 63-65 of page 3, “To date, a wide variety of important bioactivities, including antitumor, antioxidant, antimicrobial, and immunomodulatory effects, have been shown for both the polysaccharides and triterpenoids” was deleted.

14. Line 65-66 of page 3, “much attention” was revised as “much effort”.

15. Line 69-71 of page 4, “Hot water extraction (HWE) is a conventional method, and widely used in extraction of polysaccharides from mushrooms due to the fact that no special equipments are required in the extraction” was revised as “Polysaccharides are generally prepared by traditional hot water extraction due to its easy operation and low instrument input”.

16. Line 73 of page 4, “a variety of new techniques” was revised as “a variety of novel techniques”.

17. Line 75-76 of page 4, “for a higher yield and efficacy” was deleted.

18. Line 76-77 of page 4, “For extraction of triterpenoids of G. lucidum, maceration using organic extracting solvents such as ethanol is often used” was revised as “For the extraction of triterpenoids, maceration as a conventional method is adopted widely using organic solvents as an extraction solvent”.

19. Line 77-78 of page 4, “and microwave irradiation was shown to be effective for the promotion of extraction of triterpenoids from G. lucidum” was revised as “A novel technique of microwave irradiation was shown to be effective for the extraction of triterpenoids from this fungus”.

20. Line 80-82 of page 4, “To improve the extraction efficacy, the simultaneous extraction of multiple biologically active components from plant materials has become increasingly popular in recent years” was changed as “In recent years, the simultaneous extraction of multiple biologically active components from plants has become increasingly popular in order to improve extraction efficacy and reduce the number of extraction steps”.

21. Line 82 of page 4, “there have been no available reports on the simultaneous extraction of multiple objective constituents from mushrooms to date” was inserted.

22. Line 82 of page 4, “most studies” was revised as “most of previous studies”.

23. Line 84-85 of page 4, “The application of these available processes may result in the loss of either polysaccharides or triterpenoids during the extraction” was revised as “Hence, the significant loss of either polysaccharides or triterpenoids occurred during the extraction, and this would lead to a waste of resources”.

24. Line 86 of page 4, “a process for” was deleted.

25. Line 88-89 of page 4, “ultrasound technology was examined for the simultaneous extraction of polysaccharides and triterpenoids” was revised as “ultrasound technology was examined for the co-extraction of polysaccharides and triterpenoids”.

26. Line 89 of page 4, “and an optimization study is essential” was inserted.

27. Line 90-91 of page 4, “Response surface methodology (RSM) is a collection of mathematical and statistical techniques widely used for optimizing complex processes” was revised as “Response surface methodology (RSM), a collection of mathematical and statistical techniques, is widely used for optimizing complex processes”.

28. Line 93-94 of page 4-5, “RSM has been used extensively in optimizing the extraction processes of bioactive components” was changed as “RSM has been employed for optimizing the extraction of G. lucidum polysaccharides and ganoderic acids and triterpenoids from the medicinal fungus Sanghuangporus sanghuang”.

29. Line 94 of page 5, “On the other hand” was deleted.

30. Line 96 of page 5, “with different relative importances” was inserted.

31. Line 101-103 of page 5, “In an attempt to better utilize G. lucidum resources, the simultaneous extraction of polysaccharides and triterpenoids was examined using ultrasound technology in this work” was revised as “In the present study, the ultrasonic-assisted co-extraction (UACE) of polysaccharides and triterpenoids was evaluated using ultrasound technology as an attempt to better utilize G. lucidum”.

32. Line 103-104 of page 5, “The parameters of ultrasonic-assisted co-extraction (UACE) of polysaccharides and triterpenoids were optimized using RSM integrated with a desirability function” was changed as “The extraction parameters were optimized using RSM with a desirability function by setting the equal importance for the two components”.

33. Line 104 of page 5, “Furthermore” was inserted.

34. Line 107-108 of page 5, “using conventional approaches” was revised as “using conventional extraction approaches”.

35. Line 110-112 of page 5, “The present study is the first report on the simultaneous extraction of polysaccharides and triterpenoids from the medicinal mushroom G. lucidum” was deleted.

Materials and methods

1. Line 115 of page 5, “The dried G. lucidum fruiting bodies were” was revised as “The dried G. lucidum was”.

2. Line 117 of page 5, “ferric chloride, and trichloroacetic acid” was inserted.

3. Line 119 of page 6, “Conventional hot water extraction of polysaccharides” was changed as “hot water extraction of polysaccharides”.

4. Line 120 of page 6, “The G. lucidum sample was processed into a fine powder in a grinder” was revised as “The dried G. lucidum was grounded in a grinder to fine powder (40 mesh)”.

5. Line 121 of page 6, “treated” was revised as “pretreated”.

6. Line 123 of page 6, “in the matrix” was deleted.

7. Line 123-125 of page 6, “The obtained powder (1 g) was placed into a 150 mL flask containing 40 mL deionized water, and extracted at 95�C in a water bath for 2 h” was revised as “The treated powder (1 g) was mixed with 40 mL of deionized water in a 150 mL flask, and extracted for 2 h at 95�C”.

8. Line 125 of page 6, “The extraction procedure was repeated once” was inserted.

9. Line 125 of page 6, “The whole extract” was revised as “The extract”.

10. Line 126 of page 6, “The supernatant was collected, and the residue was extracted again” was deleted.

11. Line 127-128 of page 6, “The supernatants were combined, and used for the estimation of polysaccharides or for further purification before the assay of antioxidant activity” was revised as “The combined supernatants were used for the estimation of crude polysaccharides or for further purification by ethanol precipitation when used in the assay of antioxidant activity”.

12. Line 129 of page 6, “Conventional method for extraction of triterpenoids” was revised as “Ethanol maceration for extraction of triterpenoids”.

13. Line 131-132 of page 6, “The mouth of the flask was sealed with aluminum foil to avoid solvent evaporation during the extraction” was revised as “The flask was sealed with aluminum foil during the extraction”.

14. Line 132-134 of page 6, “The residue was collected after centrifugation at 8000 ×g for 10 min. The supernatant was harvested, and the residue was extracted again” was revised as “The extraction procedure was repeated once. The extraction solution was centrifuged at 8000 ×g for 10 min”.

15. Line 136-137 of page 6, “Ultrasonic-assisted co-extraction of polysaccharides and triterpenoids” was revised as “UACE of polysaccharides and triterpenoids”.

16. Line 138-139 of page 6, “An ultrasonic bath having a 3 L usable capacity was used” was revised as “The UACE was performed in an ultrasonic cleaning bath with a 3 L usable capacity”.

17. Line 139-140 of page 6, “and ultrasonic power can be set between 35 and 310 W” was deleted. 

18. Line 140-141 of page 6, “The G. lucidum fine powder (1 g) was extracted with an ethanol-water solution in a 150 mL flask” was revised as “Each of 1 g of the fine powder was extracted with an aqueous ethanol in a 150 mL flask, and the other conditions were described elsewhere”.

19. Line 141-142 of page 6-7, “The mouth of the flask was sealed with aluminum foil during the extraction” was deleted.

20. Line 143-144 of page 7, “and the resulting supernatant was used for measurements of polysaccharides and triterpenoids” was revised as “The polysaccharides and triterpenoids in the resulting supernatant were determined”.

21. Line 144-145 of page 7, “The number of extractions varied with experiments” was deleted.

22. Line 168 of page 8, “Single factor test” was revised as “Single-factor experiments”.

23. Line 169-170 of page 8, “Prior to RSM optimization, the preliminary range of process variables for the extraction was determined using one-factor-at-a-time approach with the desirability function” was revised as “The single-factor experiment was carried out to determine the preliminary range of the extraction variables”.

24. Line 170-174 of page 8, “The G. lucidum fine powder (1 g) was extracted in 150 mL flasks with varying temperatures (40 to 90�C), ethanol concentrations (20% to 80%), ultrasonic power (70 to 245 W), ratios of liquid to solid (10 to 60 mL/g), extraction times (30 to 180 min), and number of extractions (1 to 4)” was revised as “The G. lucidum fine powder (1 g) was extracted in 150 mL flasks with aqueous ethanol. The optimum extraction temperature was first determined by varying temperature from 40 to 90�C under other conditions: 40% ethanol, ultrasonic power 140 W, liquid/solid ratio 40 mL/g, extraction time 60 min, and number of extractions 1. Then, the optimum values of ethanol concentration (20%−80%), ultrasonic power (70−245 W), ratios of liquid to solid (10−60 mL/g), extraction time (30−180 min), and number of extractions (1−4) were determined sequently using determined optimal values in the extraction”.

25. Line 174 of page 8, “Each experiment was conducted in triplicate” was inserted.

26. Line 175 of page 8, “and the D values were calculated” was inserted.

27. Line 176 of page 8, “Box-Behnken design (BBD)” was changed as “Box-Behnken design (BBD) of RSM”.

28. Line 177 of page 8, “On the basis of the results of the single-factor experiments” was revised as “After the single factor experiments”.

29. Line 184-185 of page 8, “To simplify the research, one extraction number was used in all experiments” was revised as “To simplify the study, only one extraction number was tested for all experiments in this design”.

30. Line 185 of page 8, “Each trial” was revised as “Each experiment”.

31. Line 187 of page 9, “To determine the relationship” was revised as “To correlate the relationship”.

32. Line 201 of page 9, “Assay of polysaccharides and triterpenoids” was revised as “Quantitation of polysaccharides and triterpenoids”.

33. Line 206 of page 9, “at 8000 ×g for 10 min” was deleted.

34. Line 208-209 of page 10, “The vanillin-glacial acetic acid method was used for analysis of the triterpenoid amount in each sample” was revised as “The vanillin-glacial acetic acid method was used for analysis of the triterpenoid with ursolic acid as the standard”.

35. Line 210 of page 10, “and expressed as the equivalent amount of ursolic acid” was inserted.

36. Line 211 of page 10, “using D-glucose as the standard” was inserted.

37. Line 224 of page 10, “and A is the absorbance for the solution” was revised as “and A is the absorbance for the DPPH solution”.

38. Line 226 of page 10, “Reducing power assay” was inserted.

39. Line 226 of page 10, “The reducing capacity of a natural component can employ as a significant indicator of its potential antioxidant activity” was inserted.

40. Line 226 of page 10, “A higher absorbance of the reaction mixture at 700 nm shows a higher reducing power” was inserted.

41. Line 226 of page 10, “The reducing power of the polysaccharides and triterpenoids was determined following the method described by Ahmadi et al [35], with minor modifications” was inserted.

42. Line 226 of page 10, “Briefly, 2 mL sample at varying concentrations (0.1−0.5 mg/mL) was mixed with 2 mL potassium phosphate buffer (0.2 M, pH 6.6), then incubated at 50�C for 20 min. The reaction was terminated by adding 2 mL trichloroacetic acid solution (10%, w/v). Then, the reaction solution (2 mL) was mixed with 0.4 mL ferric chloride (0.1%, w/v) and distilled water (2 mL), and the absorbance was measured at 700 nm against a blank after 10 min” was inserted.

Results

1. Line 237 of page 11, “by HWE” was revised as “conventional HWE (two numbers of extraction)”.

2. Line 237-238 of page 11, “and 0.59% was obtained for the triterpenoids using the ethanol maceration” was revised as “For triterpenoids extraction using the conventional maceration method, an extraction yield of 0.59% was obtained using two extraction times”.

3. Line 240 of page 11, “on the yields of polysaccharides and triterpenoids” was added.

4. Line 241-243 of page 11, “Extraction temperature was tested between 40 and 90�C, while other variables were constant (ethanol concentration, 40%; ultrasonic power, 140 W; ratio of liquid/solid, 40 mL/g; extraction time, 60 min; and number of extractions, 1)” was deleted.

5. Line 243 of page 11, “As shown in Fig 1A” was revised as “As presented in Fig 1A”.

6. Line 244 of page 11, “within the tested range from 40 to 90�C” was inserted.

7. Line 244-246 of page 11, “The highest polysaccharide yield (0.41%) was achieved at the highest extraction temperature (90�C)” was revised as “The highest polysaccharide yield was 0.41% at 90�C”.

8. Line 246-248 of page 11, “The triterpenoid yield increased from 0.11% to 0.15% as extraction temperature changed from 40 to 80�C. When extraction temperature increased beyond 80�C, the triterpenoid yield decreased” was revised as “The triterpenoid yield increased from 0.11% to 0.15% as extraction temperature changed from 40 to 80�C, and then decreased above 80�C.”.

9. Line 248 of page 11, “As to the combined performance” was inserted.

10. Line 250 of page 11, “Above this, it started to decline” was revised as “Above this, it started to decline”.

11. Line 252 of page 12, “on the yields of polysaccharides and triterpenoids” was added.

12. Line 253-255 of page 12, “Ethanol concentrations ranging between 20%−80% were tested. Other conditions were as follows: temperature, 80�C; ultrasonic power, 140 W; ratio of liquid/solid, 40 mL/g; extraction time, 60 min; and number of extractions, 1” was deleted.

13. Line 255-257 of page 12, “Polysaccharide yield decreased in a rapid manner with increasing ethanol concentration (Fig 1B), with the maximum yield of 0.46% observed at the lowest studied concentration (20%)” was revised as “Fig 1B shows that the polysaccharide yield decreased in a rapid manner with increasing ethanol concentration in the scope of test, with the maximum yield of 0.46% observed at the lowest concentration of 20%”.

14. Line 257-258 of page 12, “the highest triterpenoid yield (0.34%)” was revised as “the highest triterpenoid yield of 0.34%”.

15. Line 260-261 of page 12, “The optimal level of ethanol was therefore around 50%” was revised as “The optimal ethanol concentration was therefore around 50%”.

16. Line 263 of page 12, “50% ethanol was used and not included in a further optimization” was revised as “the ethanol concentration of 50% was selected, and not included in BBD for further optimization”.

17. Line 264 of page 12, “on the yields of polysaccharides and triterpenoids” was inserted.

18. Line 265-268 of page 12, “The effects of ultrasonic power varying between 70 and 245 W were investigated, while other operational parameters were: extraction temperature, 80�C; ethanol concentration, 50%; ratio of liquid to solid, 40 mL/g; extraction time, 60 min; and number of extractions, 1” was deleted.

19. Line 268 of page 12, “As observed in Fig 1C” was revised as “As observed in Fig 1C”.

20. Line 271 of page 12, “applied” was inserted.

21. Line 273-274 of page 12, “An ultrasonic power of 175 W was therefore appropriate in the subsequent optimization” was changed as “An ultrasonic power of 175 W was therefore desirable in the extraction”.

22. Line 275 of page 12, “on the yields of polysaccharides and triterpenoids” was inserted.

23. Line 276-278 of page 13, “Extraction was conducted at different liquid/solid ratios from 10 to 60 mg/L under the following conditions: extraction temperature, 80�C; ethanol concentration, 50%; ultrasonic power, 175 W; extraction time, 60 min; and number of extractions, 1” was deleted.

24. Line 278-279 of page 13, “As illustrated in Fig 1D” was revised as “As presented in Fig 1D”.

25. Line 280 of page 13, “increased from” was revised as “rose from”.

26. Line286 of page 13, “As to the combin

---

## [Editor Report · Decision Letter 1]

1 Dec 2020

PONE-D-20-24854R1

Optimization of ultrasonic-assisted extraction of polysaccharides and triterpenoids from the medicinal mushroom Ganoderma lucidum and evaluation of their in vitro antioxidant capacities

PLOS ONE

Dear Dr. Liu,

Thank you for submitting your manuscript to PLOS ONE. After careful consideration, we feel that it has merit but does not fully meet PLOS ONE’s publication criteria as it currently stands. Therefore, we invite you to submit a revised version of the manuscript that addresses the points raised during the review process.

Authors have addressed all the major concerns raised in the reviewers' reports. However, one sentence in the conclusion section fails to present the verity of the study: "The antioxidant activity of the polysaccharides was enhanced by ultrasound."(L480), which does not hold the true since it means that you have somehow extracted polysaccharides, measured their antiox activity, let out the ultrasound through them and, measured their antiox activity once again and, finally, compared the two values. Please elaborate this.

Additionally, more attention should be paid to the Conclusion section .

We look forward to receiving your revised manuscript.

Kind regards,

Branislav T. Šiler, Ph.D.

Academic Editor

PLOS ONE

---

## [Author Response · Author response to Decision Letter 1]

13 Dec 2020

Responses to the reviewer’s comments

Manuscript number: PONE-D-20-24854

Title: Optimization of ultrasonic-assisted extraction of polysaccharides and triterpenoids from the medicinal mushroom Ganoderma lucidum and evaluation of their in vitro antioxidant capacities

Dear Editors,

We appreciate your comments and suggestions on our manuscript submitted previously. The manuscript has been carefully revised according to the suggestions and recommendation by the reviewers, and now all problems have been addressed in revised manuscript. Revised portion are marked in red in the paper. Point-by-point responses to these comments are showed below in this letter.

Sincerely yours,

Shi-Zhong Zheng, Wei-Rui Zhang, Sheng-Rong Liu

Reviewer 1#: Some doubts, suggestions and recommendation

Comment 1: Authors have addressed all the major concerns raised in the reviewers’ reports. However, one sentence in the conclusion section fails to present the verity of the study: “The antioxidant activity of the polysaccharides was enhanced by ultrasound”(L480), which does not hold the true since it means that you have somehow extracted polysaccharides, measured their antioxidant activity, let out the ultrasound through them and, measured their antioxidant activity once again and, finally, compared the two values. Please elaborate this.

Response: Accept. We have carefully revised this sentence. See in the revision.

Comment 2: Additionally, more attention should be paid to the Conclusion section.

Response: Accept. We have carefully revised this section, and we believe that the quality was greatly improved. See in the revision.

Comment 3: Response: Thanks. We have updated statement on the financial disclosure in our cover letter.

List of the changes in revision

Abstract

1. Line 34 of page 2, “crude” was deleted.

2. Line 35 of page 2, “crude” was deleted.

Introduction

1. Line 46 of page 3, “The most famous” was revised as “the most highly prized”.

Results

1. Line 322 of page 15, “the model was statistically very significant” was revised as “the model was statistically significant”.

2. Line 396 of page 19, “triterpenoids” was revised as “triterpenoid-rich extracts”.

Discussion

1. Line 435 of page 20, “in a convenient extraction process” was revised as “in a simple extraction process”.

2. Line 448 of page 21, “have been reported” was revised as “have been reported widely”.

3. Line 458 of page 21, “was shown” was revised as “was also shown”.

4. Line 461 of page 21, “were” was revised as “may be”.

Conclusion

1. Line 476 of page 22, “the simultaneous extraction” was revised as “the simultaneous and efficient extraction”.

2. Line 477 of page 22, “from G. lucidum” was revised as “from the medicinal fungus G. lucidum”.

3. Line 477 of page 22, “with ultrasonic technology” was revised as “with ultrasonic extraction technology”.

4. Line 480 of page 22, “On the basis of the scavenging capacity of DPPH and evaluation of reducing power” was inserted.

5. Line 480 of page 22, “The antioxidant activity of the polysaccharides was enhanced by ultrasound” was revised as “the antioxidant activity of the extracted polysaccharides by ultrasonic action in optimal UACE process was significantly enhanced”.

6. Line 480 of page 22, “while the triterpenoid-rich extracts demonstrated unchanged activity” was inserted.

---

## [Editor Report · Decision Letter 2]

16 Dec 2020

Optimization of ultrasonic-assisted extraction of polysaccharides and triterpenoids from the medicinal mushroom Ganoderma lucidum and evaluation of their in vitro antioxidant capacities

PONE-D-20-24854R2

Dear Dr. Liu,

We’re pleased to inform you that your manuscript has been judged scientifically suitable for publication and will be formally accepted for publication once it meets all outstanding technical requirements.

Kind regards,

Branislav T. Šiler, Ph.D.

Academic Editor

PLOS ONE
---

## [Editor Report · Acceptance letter]

18 Dec 2020

PONE-D-20-24854R2 

Optimization of ultrasonic-assisted extraction of polysaccharides and triterpenoids from the medicinal mushroom *Ganoderma lucidum* and evaluation of their in vitro antioxidant capacities 

Dear Dr. Liu:

I'm pleased to inform you that your manuscript has been deemed suitable for publication in PLOS ONE. Congratulations! Your manuscript is now with our production department. 

Kind regards, 

on behalf of

Dr. Branislav T. Šiler 

Academic Editor

PLOS ONE